

# pmparser and PMDB: resources for large-scale, open studies of the biomedical literature

Joshua L. Schoenbachler[1] and Jacob J. Hughey[1,2]

[1] Department of Biomedical Informatics, Vanderbilt University Medical Center, Nashville, TN, USA
[2] Department of Biological Sciences, Vanderbilt University, Nashville, TN, USA

## ABSTRACT

PubMed is an invaluable resource for the biomedical community. Although PubMed is freely available, the existing API is not designed for large-scale analyses and the XML structure of the underlying data is inconvenient for complex queries.
We developed an R package called pmparser to convert the data in PubMed to a relational database. Our implementation of the database, called PMDB, currently contains data on over 31 million PubMed Identifiers (PMIDs) and is updated regularly. Together, pmparser and PMDB can enable large-scale, reproducible, and transparent analyses of the biomedical literature. pmparser is licensed under GPL-2 and available at https://pmparser.hugheylab.org. PMDB is available in both PostgreSQL (DOI 10.5281/zenodo.4008109) and Google BigQuery (https://console.cloud.google.com/bigquery?project=pmdb-bq&d=pmdb).

## INTRODUCTION

As biomedical researchers continue to advance knowledge, the literature continues to grow. This growth, along with advances in technology, has created exciting opportunities at two levels. First, it enables biomedical discovery using techniques such as natural language processing (*Kveler et al., 2018*). Second, it enables meta-research into how research is organized, performed, disseminated, and ultimately used (*Boyack et al., 2011*; *Piwowar et al., 2018*; *Wu, Wang & Evans, 2019*; *Abdill & Blekhman, 2019*; *Hutchins et al., 2019b*; *Fu & Hughey, 2019*).

An invaluable resource for the biomedical literature is PubMed/MEDLINE, maintained by the National Library of Medicine (NLM) of the U.S. National Institutes of Health. In addition to the PubMed website, all data from PubMed are freely available through the E-utilities API and to download in bulk. However, the API is designed for specific queries or small- to moderate-scale studies, and the downloadable files store data in deeply nested XML that must first be parsed. These limitations hinder large-scale analyses of PubMed data. Other large databases such as Scopus and Web of Science are not freely available, which limits access to researchers at particular institutions and discourages reproducibility and transparency.

Corresponding author
Jacob J. Hughey,
jakejhughey@gmail.com

To address these issues, we developed two companion resources: pmparser and PMDB. pmparser is an R package allowing one to easily create and update a relational database of the data in PubMed. PMDB is our implementation of the database, which is publicly available and updated regularly.

## MATERIALS AND METHODS

The pmparser R package relies on the xml2, data.table, and DBI packages, which provide efficient parsing of XML documents into tables, manipulating the tables, sending them to a database, and manipulating the database from R. pmparser supports four database management systems: PostgreSQL, MariaDB, MySQL, and SQLite. The last is recommended only for small-scale testing.

NLM releases PubMed/MEDLINE data as a set of baseline XML files each December. Updates to the baseline, also XML files, are released daily (called post-baseline files below). Each file typically contains data on tens of thousands of PMIDs. pmparser parses the data into a set of tables organized by data type and linked by PMID (Table S1).

### Creating the database

To create the database, pmparser does the following:

1. Download the baseline XML files.
2. Initialize the tables in the database.
3. For each baseline file (in parallel) and for each data type:

   a. Parse the XML into R data.table(s).
   b. Append the data.table(s) to the corresponding table(s) in the database.

4. Add rows to the table containing the date and time at which each file was processed.
5. For each table in the database, keep only rows corresponding to the latest version of each PMID. The vast majority of PMIDs have only one version, even when their data is subsequently updated. Exceptions are articles from journals such as F1000Research.
6. Download the latest version of the NIH Open Citation Collection (*Hutchins et al., 2019a*) and add it as a table in the database.

### Updating the database

The post-baseline XML files contain data on new, updated, and deleted PMIDs. The procedure here is similar to above, except the data are first parsed into a set of temporary tables before being appended to the main tables.

1. Download the post-baseline XML files that have not yet been processed.
2. Initialize the temporary tables in the database.
3. For each post-baseline file (in parallel) and for each data type:

   a. Parse the XML into R data.table(s).
   b. Append the data.table(s) to the corresponding temporary table(s) in the database.

4. Add rows to the table containing the date and time at which each file was processed.

5. Determine the set of PMIDs whose data have been updated, along with each PMID's most recent version and the most recent XML file in which it was updated.

6. For each main-temporary table pair:

   a. Delete rows in the main table for PMIDs whose data have been updated.

   b. Append rows in the temporary table that contain each PMID's most recent data (based on version number and XML file). For instance, if a PMID is updated once in one XML file and again in a subsequent XML file, only the rows from the second update are appended. If a PMID is marked as deleted in an XML file, then that PMID's rows are deleted from the main table, but no rows are appended.

7. Delete the temporary tables.

8. If a newer version of the NIH Open Citation Collection exists, download it and overwrite the corresponding table in the database.

## RESULTS

Using pmparser, we created our implementation of the database, PMDB, from the baseline XML files and the latest version of the NIH Open Citation Collection. Creating the database in PostgreSQL using 24 cores on a computer with 128 GiB of memory took 5.6 h. Updating the database from the seven post-baseline XML files released between December 14, 2020 and December 17, 2020 took another 20.8 min. Step-by-step instructions for using the PostgreSQL version of PMDB are available on Zenodo (DOI 10.5281/zenodo. 4008109). The most recent compressed dump is 20.8 GiB and includes data for 31,872,359 PMIDs (Table S2).

To make PMDB more widely accessible, we also made it publicly available on Google BigQuery (https://console.cloud.google.com/bigquery?project=pmdb-bq&d=pmdb), which is designed for large-scale and high-speed analytics. With BigQuery, users can access PMDB without having to set up their own database server, and are responsible only for the costs of their queries. We plan to update both versions of PMDB monthly.

To demonstrate PMDB's utility and scale, we performed two analyses as examples of possible use cases. First, we used the pub_history and author tables to quantify the increase in the number of authors per published article over time (Fig. 1). Second, we used the pub_history, author, and author_identifier tables to quantify the increase in the fraction of author names with an ORCID identifier over time (Fig. 2). Because these analyses involve multiple data types and millions of PMIDs, performing them using the E-utilities API would be difficult and time-consuming. In contrast, using PMDB makes them straightforward (~10 lines of SQL) and fast (~1 min). Code to reproduce our results using the Postgres and BigQuery versions of PMDB is available on Zenodo (DOI 10.5281/ zenodo.4004909).
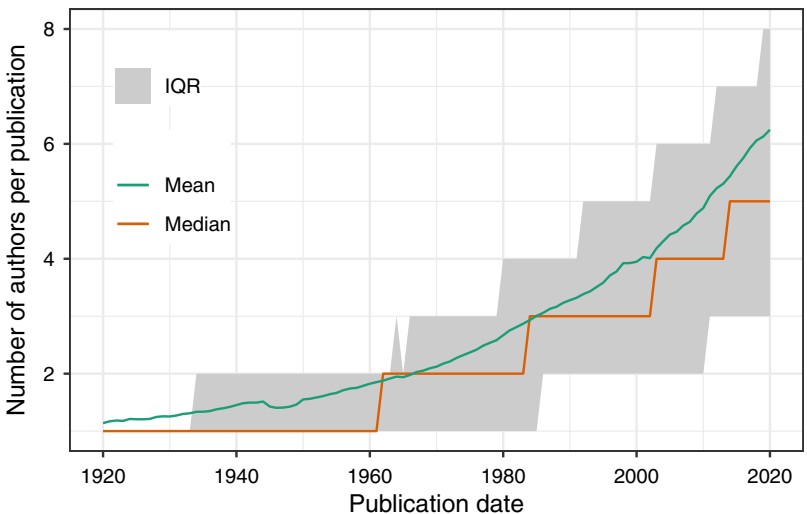

**Figure 1 Using PMDB to quantify the number of authors per publication between 1920 and 2020, grouped by year.** The database query involved joining the pub_history and author tables on the pmid field. The MEDLINE XML documentation states that for PMIDs created between 1984 and 1995, at most 10 authors were entered, and for PMIDs between 1996 and 1999, at most 25 authors were entered. These limits do not affect the median or interquartile range, but may introduce inaccuracies to the mean.

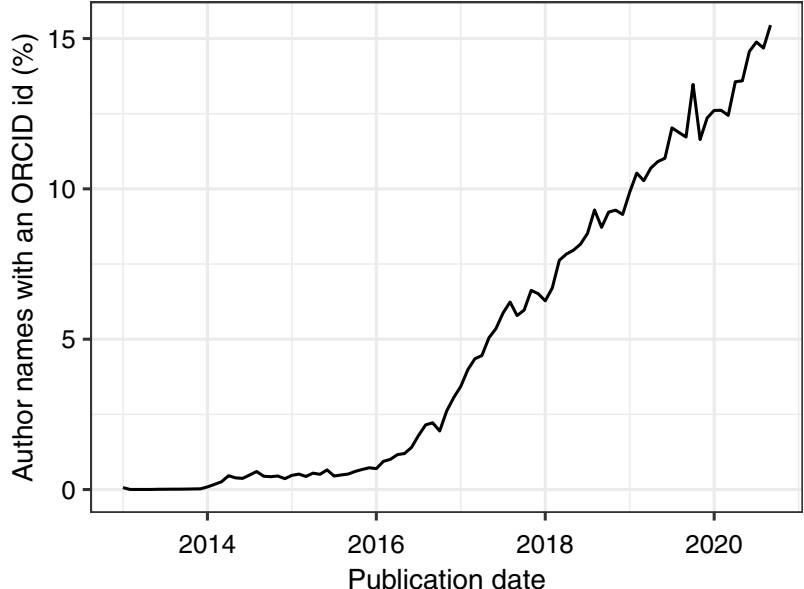

**Figure 2 Using PMDB to quantify the percentage of author names with an ORCID identifier from January 2013 to August 2020, grouped by month.** The database query involved joining the pub_history, author, and author_identifier tables on the pmid and author_pos fields. ORCID identifiers became available in October 2012.

## DISCUSSION

pmparser offers similar functionality to a Python package called medic (https://github.com/fnl/medic), which is no longer maintained. The main differences between medic and pmparser are that the latter parses considerably more of the MEDLINE XML

(e.g., author affiliations and identifiers and everything related to investigators), smoothly handles PMID versions, and makes it easier to update the database.

pmparser also offers similar functionality to a recently developed Python package called pubmed_parser (*Achakulvisut, Acuna & Kording, 2020*), with several differences. First, the packages differ somewhat in which elements of the XML they parse. For example, pmparser parses the History section, which contains data on when an article was submitted, accepted, etc., but pubmed_parser does not. Second, although both packages parse the abstracts (a common data source for biomedical NLP), pubmed_parser can also parse the full text of articles in the PubMed Central Open Access Subset. Third, pmparser adds each XML file's data to tables in a relational database, whereas pubmed_parser outputs the data as a list of Python dictionaries. Finally, pmparser can update an existing database, whereas pubmed_parser is designed to process an entire corpus at once. Overall, which package is more suitable will depend on the use case. We anticipate, however, that most researchers will prefer to bypass these packages and simply use PMDB in Postgres or BigQuery.

As our examples highlight, PMDB's structure offers great flexibility to query and join multiple data types. What are the most common Medical Subject Heading (MeSH) terms associated with articles published by a given author? What are the datasets linked to articles published by authors with a given institutional affiliation? What are the articles published in the last five years in a given journal that have cited a given article? Questions such as these are now easily answerable using PMDB. Furthermore, in contrast to analyses based on subscription databases, analyses based on PMDB can be released without restriction for the entire community to verify and build on.

With pmparser and PMDB, researchers seeking to use the wealth of data in PubMed have two new options: create their own implementation of the database or use ours. Either way, once the database is ready, users can query it directly or pull the data into their tool of choice for statistical analysis, machine learning, etc. Together, pmparser and PMDB can enable large-scale, reproducible, and transparent analyses of the biomedical literature.

## ACKNOWLEDGEMENTS

We thank Elliot Outland for contributing to pmparser. We thank Darwin Fu for helpful comments on the manuscript.

### Funding

This work was supported by the National Institutes of Health (R35GM124685).
The funders had no role in study design, data collection and analysis, decision to publish, or preparation of the manuscript.

## Grant Disclosures

The following grant information was disclosed by the authors:
National Institutes of Health: R35GM124685.

## Competing Interests

Jacob J. Hughey is an Academic Editor for PeerJ.

## Author Contributions

- Joshua L. Schoenbachler conceived and designed the experiments, performed the experiments, analyzed the data, authored or reviewed drafts of the paper, and approved the final draft.
- Jacob J. Hughey conceived and designed the experiments, performed the experiments, analyzed the data, prepared figures and/or tables, authored or reviewed drafts of the paper, and approved the final draft.

## Data Availability

pmparser is available at https://pmparser.hugheylab.org.

PMDB is available at Google BigQuery (https://console.cloud.google.com/bigquery?project=pmdb-bq&d=pmdb) and Zenodo:

Hughey, Jacob, & Schoenbachler, Joshua. (2021). PMDB: a relational database for PubMed [Data set]. Zenodo. DOI 10.5281/zenodo.4499311.

Code to reproduce our results is also available at Zenodo:

Hughey, Jacob, & Schoenbachler, Joshua. (2020). Reproducible results for: pmparser and PMDB: resources for large-scale, open studies of the biomedical literature [Data set]. Zenodo. DOI 10.5281/zenodo.4422037.

## Supplemental Information

Supplemental information for this article can be found online at http://dx.doi.org/10.7717/peerj.11071#supplemental-information.

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
