# Peer review of "pmparser and PMDB: resources for large-scale, open studies of the biomedical literature"

_PeerJ, doi:10.7717/peerj.11071_

## Round 0.1 · original submission · Major Revisions

The two reviewers have raised the major concerns regarding the manuscript.

Specific comments:

1) A number of use cases should be provided to demonstrate the usefulness of this tool.
2) The performance of the tool should be evaluated in order to compare with other existing tools.
3) More work should be done on the figures and explanation and clarification should be strengthened.
4) A lot of details such as database tables and fields should be added.

·

Basic reporting

Thank you very much for an excellent manuscript describing this useful tool. The language is clear and the reporting is concise.

General comments

The quality of the tool is well described, and as an advanced user of PubMed for text mining applications, the value is also clear. However, I worry that "casual" readers may be a little unsure of the utility, as you offer no use cases. I don't think is necessary to carry out advanced analyses for this short report, but a discussion (beyond the very brief overview in the introduction) of some of the analyses that this tool facilitates would be extremely useful context.

On a related note, I find figures 1 and 2 and the results they show to be a little out of place. They do not seem to be related to the discussion of the tool and I am not convinced that these are relevant results. Is the increasing number of authors relevant to the complexity of the database? Is the relatively low number of ORCID authors obstructing analyses using ORCID?

Table S1 provides an overview of the fields in the database with a very sparse descriptor in the second column. This should be elaborated. Is there any information beyond the metadata listed in table S1? Could one extract citation structures? Co-author structures? Please clarify.

For the uninitiated, it would also be useful with a brief description or recommendation for accessing and analyzing the data in SQL format. I imagine many user will expect either a point-and-click style analysis option, which is of course not an option here (nor is it the purpose).

Minor comments

Line 22: "As biomedical researchers continue to push into the unknown" I think I know what you are trying to say here, but I think this sentence is a bit unclear. What does it mean to "push into the unknown"? I would exactly consider that part of my job description!

Line 29: "The definitive resource for the biomedical literature is PubMed/MEDLINE". I'm not so sure about this. Google Scholar comes to mind as a strong competitor. Perhaps find a different word that "definitive" - no one knows what will happen in 10 years.

Experimental design

No comment.

Validity of the findings

No comment.

Reviewer 2 ·

Basic reporting

The manuscript by Schoenbachler and Hughey describes a straightforward method of converting the data in PubMed from its native XML format to a single relational database and augmenting the PubMed data with citation information from the NIH Open Citation Collection (iCite), updated monthly. While all of the data is already publicly available, having a regularly updated data stream that links these data sources may lower the barrier of entry for researchers lacking the resources to process the XML themselves. The manuscript itself is direct, well-written, and adheres to PeerJ standards for article organization.

That said, the authors should address the following issues:

1. Major issue: The authors are using only some of the citation information from snapshots of the iCite database (according to their code: get_citation.R) but have not included other metrics and metadata, including Relative Citation Ratio and Approximate Potential to Translate. For maximum utility to researchers, it would be beneficial to include the full set of iCite fields in the pipeline.
2. Minor issue: The authors compare their work to another PubMed parsing project, pubmed_parser, with similar goals and a largely overlapping approach. The authors detail the some of the differences between the projects, but more elaboration would be helpful for understanding when pmparser would be the better option. Additionally, a search for other PubMed database mirrors returned a different python package called “medic” (https://github.com/fnl/medic), which hasn’t been updated for a few years but claims many of the same features as pmparser, including relational database storage and incremental updates. How does it compare to pmparser?

Experimental design

As a project meant to increase the accessibility of biomedical citation information, this manuscript is within the scope of PeerJ’s focus. The experimental design and methods underlying the overall pipeline are described sufficiently in the methods section, however additional description of how the two figures were generated is needed. The first figure needs more clarification that the MEDLINE author limits may be affecting the metrics, and that the results shown may not be an accurate representation of the maximum author count. The second figure appears to be incorrectly labeled: it is not reporting the percentage of authors with an ORCID identifier, but rather the percentage of author names, without the deduplication or disambiguation that would be required to extract unique authors. The methodology behind these figures, including the specific database fields used to generate them, should be better explained in the legend or text.

Validity of the findings

Overall, the pmparser pipeline and code described in this manuscript are well documented. However, the supplemental tables need more information to be useful. The authors present a list of tables and fields stored in the database, but no description of what those fields represent. At a minimum, a description of each of the database tables and data fields are needed to help users make sense of the wealth of analytical options available to them.

---

## Round 0.2 · accepted · Accept

Thanks for the revision. I am happy to accept the current version of the manuscript for publication.

Reviewer 2 ·

Basic reporting

no comment

Experimental design

no comment

Validity of the findings

no comment

Additional comments

The authors have adequately addressed reviewer comments in this revision.